# PERFOGRAPH: A Numerical Aware Program Graph Representation for Performance Optimization and Program Analysis

**Ali TehraniJamsaz[1][*], Quazi Ishtiaque Mahmud[1][*], Le Chen[1], Nesreen K. Ahmed[2], Ali Jannesari[1]**

[1] *Iowa State University, Ames, Iowa, USA*
{tehrani, mahmud, lechen, jannesari}@iastate.edu
[2] *Intel Labs, Santa Clara, CA, USA*
nesreen.k.ahmed@intel.com

## Abstract

The remarkable growth and significant success of machine learning have expanded its applications into programming languages and program analysis. However, a key challenge in adopting the latest machine learning methods is the representation of programming languages, which directly impacts the ability of machine learning methods to reason about programs. The absence of numerical awareness, aggregate data structure information, and improper way of presenting variables in previous representation works have limited their performances. To overcome the limitations and challenges of current program representations, we propose a graph-based program representation called PERFOGRAPH. PERFOGRAPH can capture numerical information and the aggregate data structure by introducing new nodes and edges. Furthermore, we propose an adapted embedding method to incorporate numerical awareness. These enhancements make PERFOGRAPH a highly flexible and scalable representation that effectively captures programs' intricate dependencies and semantics. Consequently, it serves as a powerful tool for various applications such as program analysis, performance optimization, and parallelism discovery. Our experimental results demonstrate that PERFOGRAPH outperforms existing representations and sets new state-of-the-art results by reducing the error rate by 7.4% (AMD dataset) and 10% (NVIDIA dataset) in the well-known Device Mapping challenge. It also sets new state-of-the-art results in various performance optimization tasks like Parallelism Discovery and NUMA and Prefetchers Configuration prediction.

## 1 Introduction

In recent years, the remarkable success of machine learning has led to transformative advancements across numerous fields, including compiler optimization and program analysis. The applications include compiler heuristics prediction, optimization decisions, parallelism detection, etc. [4, 15]. The training process generally involves feeding program data as input and transforming it into a representation suitable for machine learning models. The selection of program representation is crucial, as it can significantly impact the performance of the machine learning model [12]. With the development of graph neural networks (GNNs), an increasing number of graph representations of programs have been incorporated into GNN models for program analysis [3, 28, 10]. One of the pioneering efforts in developing a comprehensive graph representation for programs is PROGRAML [14]. PROGRAML incorporates control, data, and call dependencies as integral components of a program's representation. In contrast to prior sequential learning systems for code, PROGRAML

---

[*]Equal contribution.

37th Conference on Neural Information Processing Systems (NeurIPS 2023).

closely resembles the intermediate representations used by compilers, and the propagation of information through these graphs mimics the behavior of typical iterative data-flow analyses. Despite the success that PROGRAML has achieved, there are shortcomings in this current state-of-the-art program representation, especially for performance-oriented downstream tasks. These limitations stem from neglecting numerical values available at compile time and the inadequate representation of aggregate data types.

In this paper, we present PERFOGRAPH to address the limitations of the current state-of-the-art program representation. Additionally, we propose a novel way to embed numbers in programs in an elegant way so that our DL model will not face unknown numbers during inference time. Our experiments demonstrate that PERFOGRAPH sets new state-of-the-art results in numerous downstream tasks. For example, in the Device Mapping downstream task, PERFOGRAPH yield error rates as low as 6% and 10% depending on the target hardware. Moreover, PERFOGRAPH even outperforms the tools and models specially designed for specific tasks such as parallelism discovery.

Overall, the main contributions of this paper are:

- An enhanced compiler and language-agnostic program representation based on PROGRAML that represents programs as graphs.
- The proposed representation supports aggregate data types and provides numerical awareness, making it highly effective for performance optimization tasks.
- Evaluation of the proposed representation on common downstream tasks and exceeding the performance of PROGRAML.
- Quantification of the proposed approach on a new set of downstream tasks such as parallelism discovery and configuration of NUMA systems.

The rest of the paper is structured as follows: Section 2 presents the related works. In section 3, we provide a motivational example, showing the limitations of PROGRAML, the state-of-the-art program representation. This section is followed by section 4 where we present our proposed representation PERFOGRAPH along with the novel way of embedding numerical values. In section 5, experimental results on downstream tasks are provided, and finally, Section 6 concludes the paper and discusses some future works.

## 2 Related Works

Machine learning has brought significant advancements in many fields, and program analysis and software engineering are no exceptions. However, Machine Learning (ML) and Deep Learning (DL) models can not directly process raw source code to reason about programs. Therefore, researchers have explored different approaches to represent applications in a format suitable for DL models. Generally, there are three types of commonly used program presentations: sequence of tokens, Abstract Syntax Tree (AST), and Intermediate Representation (IR).

**Sequence of tokens:** The initial attempts [16, 21, 27] represented source code as a sequence of tokens, such as identifiers, variable names, or operators. This approach intuitively treats programming languages similarly to natural languages. It allows for the utilization of advanced natural language process (NLP) techniques, such as large language models [19, 22, 23]. However, this token-based representation overlooks the inherent dependency information within the program's structure. It fails to capture the unique relationships and dependencies between different elements of the code, which can limit its effectiveness in tasks such as compiler optimization and code optimization.

**AST:** An AST represents the structure of a program by capturing its hierarchical organization. It is constructed based on the syntactic rules of the programming language and provides a high-level abstraction of the code. Previous works have leveraged ASTs as inputs to tree-based models for various code analysis tasks like software defect prediction [17] and code semantic study [9]. Moreover, there have been efforts to augment ASTs into graphs that incorporate program analysis flows such as control flow and data flow. These AST-based graph representations capture more comprehensive code dependency information and have shown superior results compared to traditional approaches in previous works [2, 3].

**IR:** IR is an intermediate step between the source code and the machine code generated by a compiler. Previous work [39] has utilized IR to train an encoding infrastructure for representing programs as a distributed embedding in continuous space. It augments the Symbolic encodings with the flow of information to capture the syntax as well as the semantics of the input programs. However, it

generates embedding at the program or function level and also requires a data-flow analysis type for generating the embedding. In contrast, our approach derives embedding from the representation and works at the more fine-grained instruction level. More recent works [6, 14, 8] have leveraged IR-based graph representation to better capture essential program information, such as control flow, data flow, and dependencies. However, despite their success, IR-based graph representations have certain limitations. For example, these representations may not be numeric-aware or may lack the ability to adequately represent aggregate data types. In this work, we propose PERFOGRAPH, a graph representation based on IR, to address these limitations.

## 3  Motivation

As stated in the related work section, program representations based on the intermediate representation are very effective in enabling DL models to automate the process of various optimizations. One such representation is PROGRAML, whose performance surpasses other code representations, making it state-of-the-art in various optimizations and downstream tasks. However, despite its potential, it suffers from several limitations. To name a few: it is incapable of properly carrying out information regarding read and write operations to the memory location, has no support for aggregate data types, and discards numerical values. Listing 1 shows a code snippet where a 3-dimensional array is defined. Figure 1 shows the PROGRAML representation of this code snippet. For illustration purposes, instruction nodes and control flow edges are shown in blue, whereas red represents variable, constant nodes, and data flow edges. Green edges show the call graph. As it can be seen, PROGRAML fails to represent some critical information. For instance, code `float arr[2][3][4]` is converted to LLVM IR `[2 x [3 x [4 x float]]]*`, which is used to construct a node in PROGRAML. It eliminates the aggregate data structure information, like the array's dimension. Leaving it up to the DL model to infer the meaning behind the numbers in `[2 x [3 x [4 x float]]]*`. Moreover, in this representation, only the type of numbers (e.g., `int8`, `float`) are considered, and the actual values of the numbers are not given attention. The absence of numerical awareness limits the performance of PROGRAML in applications where numerical values play an important role. A numerically aware representation can help understand and optimize operations involving numeric data types, constants, and expressions. There are also some anomalies in the way temporary variables are depicted. For example, in 1, we see the fourth `alloca` node allocates memory for a variable, and two `store` instructions are applied on two separate nodes representing the variable. Thus, the information about the first `store` instruction is not carried out properly when the second `store` instruction happens. In the following section, we will see how PERFOGRAPH effectively addresses many limitations in the current state-of-the-art program representation. PERFOGRAPH uses PROGRAML representation as its initial graph and reconstructs the graphs by addressing the aforementioned limitations.

```
1   #include <stdio.h>
2
3   int main(int argc, char *argv[]) {
4     int arr_state = 0;
5     float arr[2][3][4]
6       = {{{1.0f,1.0f,1.0f,1.0f},
7           {2.0f,2.0f,2.0f,2.0f},
8           {3.0f,3.0f,3.0f,3.0f}},
9          {{4.0f,4.0f,4.0f,4.0f},
10          {5.0f,5.0f,5.0f,5.0f},
11          {6.0f,6.0f,6.0f,6.0f}}};
12      arr_state = 1;
13      return 0;
14  }
```

Listing 1: C++ code example.

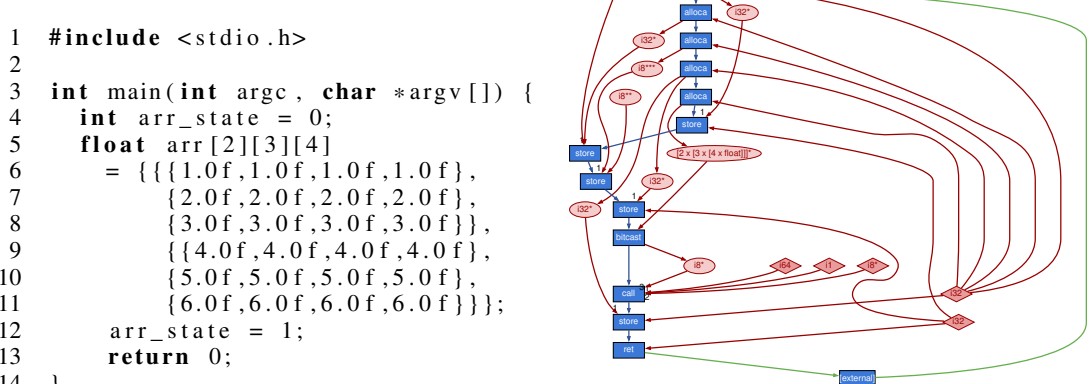

Figure 1: ProGraML representation of Listing 1.

## 4  PERFOGRAPH: A fined-grained numerical aware graph representation

PERFOGRAPH is graph representation based on LLVM IR. It is built on top of PROGRAML; however, it does not suffer from the limitations that the PROGRAML has, helping DL models to reason over the complex structure of the programs and enabling them to make more accurate optimization decisions,

especially in terms of performance optimization. Figure 2 shows how various enhancements and improvements are applied to construct a more precise representation. Consider a simple code example of defining a variable and increasing it by one {int i = 0; i++;}. Figure 2a shows the PROGRAML representation of this code example. It has two store nodes, as one is responsible for storing 0 and the other one storing the incremented value of i. In the following subsection, we will explain how PERFOGRAPH is constructed by addressing the limitations shown in Figure 2a.

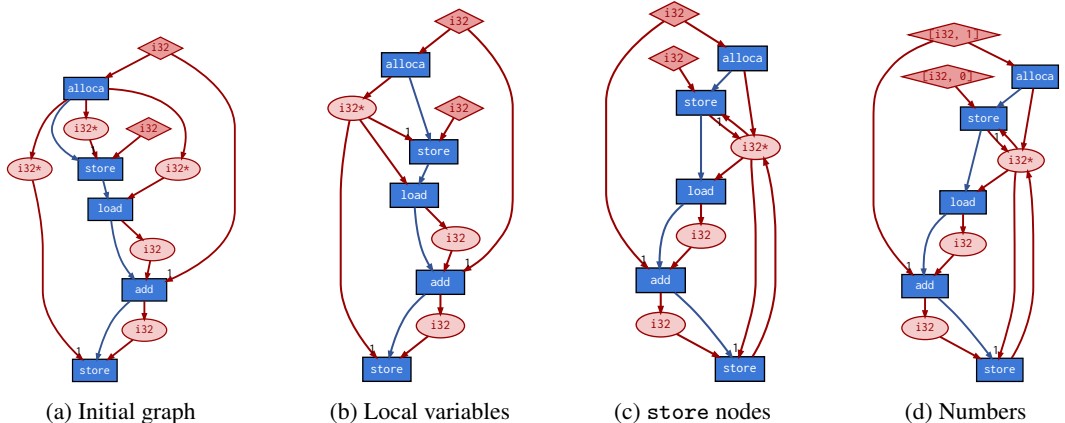

(a) Initial graph      (b) Local variables      (c) store nodes      (d) Numbers

Figure 2: PERFOGRAPH addresses the existing limitations in program representation.

## 4.1 Representing Local Identifiers and store instruction

**Local Identifiers:** Local identifiers' names are preceded by % in LLVM Intermediate representation. Memory allocation on the stack is done by alloca instruction. One of the limitations of the current state-of-the-art program representation, PROGRAML, is that it is unable to carry out information regarding the operations that happen to a memory location. For instance, in Figure 2a, the two store nodes represent storing values of 0 and 1 to variable i. However, as shown, each store instruction node is connected to a separate variable node, making it difficult for the graph neural network to reason over the operations that happen to a memory location. For the embedding vector of the second store node in 2a to represent the fact that some information regarding the variable i has changed, one has to increase the number of GNN layers to 3 to support up to 3 hops when propagating the messages in GNN. This can potentially limit the ability of the GNN model if there are a greater number of hops between the two store nodes shown in Figure 2a. To address this limitation, instead of having more than one variable node (oval-shape nodes) per identifier, PERFOGRAPH only considers one variable node in its graph representation. Any load or store instruction will refer to the same variable node. These changes are shown in Figure 2b. We see that the store nodes in Figure 2b access the same memory location, thus representing the fact that those store instructions are modifying the same memory location.

**Store instruction:** LLVM uses store instruction to write into memory. store instruction has two arguments: a value to store and the address to which it will store the value. PROGRAML differentiates between these two arguments by adding a position feature to the edges as shown in Figure 2a. However, since the store instruction modifies the contents at the corresponding memory address, we posit that it is better to reflect the fact the content of the identifier has changed. To present this information, PERFOGRAPH adds an extra edge from the store node to node representing the identifier whose value is modified by the store instruction. Figure 2c shows these changes in the graph constructed by PERFOGRAPH.

**Numbers:** Numbers can be a significant factor in optimization decisions. For example, they can show the loop bound, and different optimizations can be considered depending on the loop bound. PERFOGRAPH, unlike PROGRAML, not only considers the type of numbers such as i32, i64, float but also the actual values of the numbers. As illustrated in Figure 2d, numerical constant nodes have the actual value of the number in their feature set in addition to the type of the number. Even though numerical constant nodes have the value of the number as one of their features, there is a need to embed the numbers in a way that unknown numbers will not be seen in the inference. Unlike other tokens, numbers are harder to embed as an infinite amount of numbers exists, and to handle all ranges of numbers, we need to have a very large vocabulary set.

## 4.2 Numerical Awareness

However, with a very large vocabulary size, the DL models may still encounter numbers in the inference phase that they have not seen in the training phase. We propose a novel way of embedding numbers called Digit Embedding. Figure 3 shows our approach. To embed a number, we first break down the number to its digits; then, we consider a position for each one of the digits. The goal is to let DL models realize the place value of each digit. Then, each digit and its corresponding position are embedded and summed together. Therefore, we will have an embedding representing the information about the digits and their positions. For instance, in Figure 3, we embed each digit and its corresponding position with an output dimension of 3. Since the number has four digits, the results would be a vector/tensor of size $4 \times 3$. To make sure the Digit Embedding of numbers has the same length across numbers

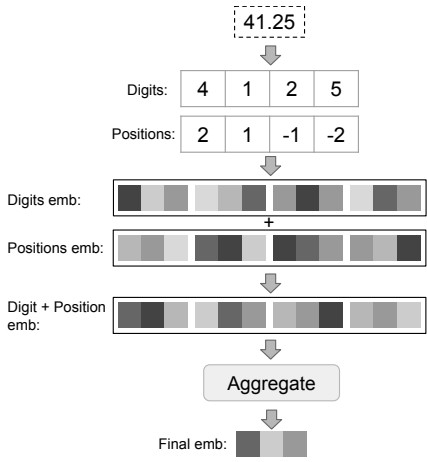

Figure 3: Overview of the digit embedding.

with varying sizes of digits, we apply an aggregation function over the embedding dimension. Since the output embedding dimension is three in this example, we would have one vector of length three representing the number after aggregation. The aggregation function can be of any type (Max, Mean, etc.).

## 4.3 Aggregate Data Types

Aggregate data types, such as arrays and vectors, are an essential part of applications. They play an important role in many applications, such as matrix multiplications. Thus, presenting these data types helps the DL models better understand programs. Current LLVM IR-based program representations fail to present aggregate data types appropriately. For example, consider a three-dimensional integer array. In LLVM IR, this array is shown as `3 x [2 x [3 x i32]]]*`. As can be seen, the length of the arrays and their data types are inferable. However, without proper representation, the DL model's capacities will be spent on learning these deterministic facts (i.e., the length of the arrays and their type). PERFOGRAPH considers aggregate data types as a new node type in its representations. Figure 4b shows how aggregate data types are supported by PERFOGRAPH. Unlike other LLVM IR-based representations, PERFOGRAPH supports multi-dimensional arrays and vectors. PERFOGRAPH creates a chain of nodes to present the different dimensions of the arrays. In Figure 4a, we see there is a node representing the three-dimensional array `[3 x [2 x [3 x i32]]]*`. PERFOGRAPH breaks down the corresponding node into three (since it is a three-dimensional array) white nodes as shown in Figure 4b. Then, each node has a context representing that specific dimension of the array. For example, the context for the third dimension is `[3 x i32]`, whereas for the second dimension, the context is `[2 x [3 x i32]]`. For each aggregate type node, in addition to the context of the node,

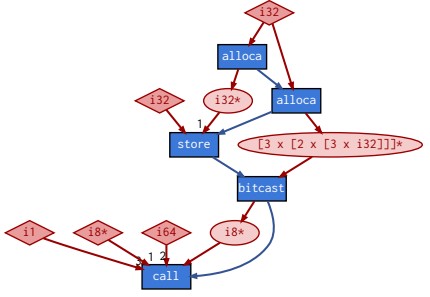

(a) No support for aggregate data types.

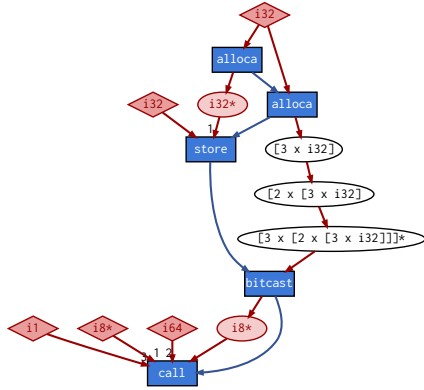

(b) Break-down of aggregate data types.

Figure 4: PERFOGRAPH supports aggregate data types.

we specifically add the length of the array and its type as additional features. For aggregate data types whose lengths are not known during compile time, we follow the LLVM conventions by considering the length of those data types as `vscale`. These enhancements will help the DL models to reason over the dimensions and types of aggregate data types. As a result, PERFOGRAPH will ultimately enable the DL models to have more accurate predictions for applications that deal with arrays and vectors.

# 5 Experimental Results and Downstream Tasks

In this section, we evaluate PERFOGRAPH on six downstream tasks. For each downstream task, we will explain the task itself, the dataset, and the baselines.

## 5.1 Experimental Setup

In our experiments, we use DGL's [40] RGCN [37] implementation for PERFOGRAPH representation. The graphs from PERFOGRAPH are treated as heterogeneous and managed via the `HeteroGraphConv` module. We use a hardware setup of two 18-core Intel Skylake 6140 CPUs and two NVIDIA Tesla V100-32GB GPUs. The embedding space for numbers is generated by extracting digits and positions from a numeric token of an IR statement, then passed to a PyTorch [31] embedding layer for digit and position embeddings. These are combined for the final numeric token embedding. Non-numeric tokens directly go through the PyTorch embedding layer. Each PERFOGRAPH heterogeneous node converts to a 120-dimensional vector via this embedding. We use the `Adam` [26] Optimizer, `relu` [1] activation function, a learning rate of $0.01$, and `hidden_dim` parameter of $60$. Mean aggregation is applied to combine different node-type results before a linear classification layer, which outputs a probability distribution for each class. The class with the highest probability is the prediction.

## 5.2 Device Mapping

**Problem Definition:** We apply PERFOGRAPH to the challenging heterogeneous device mapping [13] problem. In this task, there are a number of OpenCL kernels that we need to predict which accelerator (CPU or GPU) yields higher performance. We compare PERFOGRAPH against DeepTune [13], Inst2Vec [6], and PROGRAML [14]. The results of the baselines are quoted from [14].

**Dataset:** For this task, we use the dataset published in [13]. In this dataset, there are 256 OpenCL kernels available, and 680 LLVM IR instances are extracted from them. There are

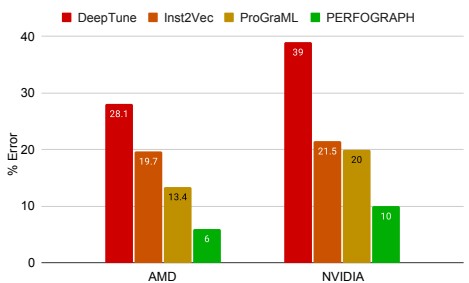

Figure 5: Performance comparison the device mapping task with state-of-the-art models [lower is better].

two types of GPUs: AMD and NVIDIA. For each of the GPUs, the runtimes of the kernels are recorded in the dataset. For AMD, 276 kernels show better performance in GPU, while 395 kernels show better performance in CPU. Whereas for NVIDIA, 385 kernels have better runtimes with GPU, and 286 kernels have better runtimes with CPU. We consider this as a binary CPU or GPU classification problem.

**Results:** As the dataset is small, we use the same 10-fold validation (with 80% training, 10% validation, and 10% testing) like PROGRAML [14] and chose the model with the highest validation accuracy. The hand-crafted features of [20] are also used as graph-level features in our model to enhance the performance following the approach in [14]. Table 1 and 2 show the final precision, call, f1-score, and accuracy for AMD and NVIDIA devices. Figure 5 compares PERFOGRAPH with state-of-the-art models on the Device Mapping dataset. We can see that PERFOGRAPH sets new state-of-the-art results by achieving the lowest error rate among the baselines both for AMD and NVIDIA, indicating the effectiveness of PERFOGRAPH.

Table 1: PERFOGRAPH results for AMD devices.

|  | Precision | Recall | F1-score | Accuracy |
|---|---|---|---|---|
| CPU | 0.94 | 0.94 | 0.94 | 0.94 |
| GPU | 0.94 | 0.94 | 0.94 | |

Table 2: PERFOGRAPH results for NVIDIA devices.

|  | Precision | Recall | F1-score | Accuracy |
|---|---|---|---|---|
| CPU | 0.87 | 0.90 | 0.89 | 0.90 |
| GPU | 0.92 | 0.89 | 0.90 | |

## 5.3 Parallelism Discovery

**Problem Definition:** In this problem, given a sequential loop, we try to predict whether a loop can be executed in parallel. We treat this problem as a binary classification problem with two classes: Parallel and Non-Parallel.

**Dataset:** The OMP_Serial dataset [11] is used for this task. It contains around 6k compilable source C files with Parallel and Non-Parallel loops. The training dataset contains around 30k IR files. The OMP_Serial dataset has three test subsets to compare the performance with three traditional parallelism assistant tools: Pluto (4032 IR files), AutoPar (3356 IR files), and DiscoPoP (1226 IR files).

**Results:** We evaluate PERFOGRAPH on all three subsets and compare it with traditional rule-based tools: Pluto [7], AutoPar [33], DiscoPoP [29], and also Deep Learning based tools: Graph2Par [11], PROGRAML. Table 3 shows the results. The results of Pluto and Graph2par are reported from [11]. As PROGRAML does not have this downstream task in their paper, we used the PROGRAML representation in our pipeline to generate the results. Results show that traditional rule-based tools have the highest precision but the lowest accuracy because those tools are overly conservative while predicting parallel loops. So, they miss out on a lot of parallelism opportunities. PERFOGRAPH achieves considerably good precision scores across all the test subsets. In terms of accuracy, PERFO-GRAPH surpasses the current state-of-the-art approaches by 2% in the Pluto and AutoPar subset. In the DiscoPoP subset, it achieves an impressive 99% accuracy and surpasses PROGRAML by 9%.

Table 3: Performance comparison of PERFOGRAPH on the OMP_Serial dataset.

| Subset | Approach | Precision | Recall | F1-score | Accuracy |
|---|---|---|---|---|---|
| | Pluto | 1 | 0.39 | 0.56 | 0.39 |
| Pluto | Graph2par | 0.88 | 0.93 | 0.91 | 0.86 |
| | PROGRAML | 0.88 | 0.88 | 0.87 | 0.89 |
| | **PERFOGRAPH** | 0.91 | 0.90 | 0.89 | **0.91** |
| | AutoPar | 1 | 0.14 | 0.25 | 0.38 |
| autoPar | Graph2par | 0.90 | 0.79 | 0.84 | 0.80 |
| | PROGRAML | 0.92 | 0.69 | 0.67 | 0.84 |
| | **PERFOGRAPH** | 0.85 | 0.91 | 0.85 | **0.86** |
| | DiscoPoP | 1 | 0.54 | 0.70 | 0.63 |
| DiscoPoP | Graph2par | 0.90 | 0.79 | 0.84 | 0.81 |
| | PROGRAML | 0.92 | 0.94 | 0.92 | 0.91 |
| | **PERFOGRAPH** | 0.99 | 1 | 0.99 | **0.99** |

## 5.4 Parallel Pattern Detection

**Problem Definition:** Parallel loops often follow some specific patterns. Identifying parallel patterns is important because it helps developers understand how to parallelize a specific program since each parallel pattern needs to be treated differently. As a result, we apply PERFOGRAPH to identify potential parallel patterns in sequentially written programs. Only the three most common parallel patterns are considered: Do-all (Private), Reduction, and Stencil [34]. Given a loop, the task is to predict the pattern.

**Dataset:** For this experiment, we also use the OMP_Serial dataset [11]. This dataset contains source codes of different parallel patterns. These programs are collected from well-known benchmarks like NAS Parallel Benchmark [24], PolyBench [32], BOTS benchmark [18], and the Starbench benchmark [5]. Then, template programming packages like Jinja [35] are used to create synthetic programs from the templates collected from the mentioned benchmarks. The dataset contains 200 Do-all (Private), 200 Reduction, and 300 Stencil loops.

**Results:** We used 80% of the dataset for training and 20% for testing. Table 4 represents our findings. The results of Pragformer [25] and Graph2par [11] are reported from [11]. We compare with these two approaches as they are specifically developed for solving this problem. For generating the results with PROGRAML, we used the PROGRAML representation in our pipeline. We can see PERFOGRAPH achieves an impressive 99% accuracy on the OMP_Serial Parallel Pattern dataset. It surpasses the state-of-the-art PROGRAML model by 3%. This indicates the strength of PERFOGRAPH to capture the syntactic and structural patterns embedded into source programs. From Table 4, we can also see PERFOGRAPH has high precision for all three patterns and achieves a high precision score for Do-all and Stencil patterns while maintaining very good accuracy.

Table 4: Performance comparison for the parallel pattern detection task with PERFOGRAPH on the OMP_Serial Dataset.

| Approach | Pattern | Precision | Recall | F1-score | Accuracy |
|---|---|---|---|---|---|
| Pragformer | Do-all | 0.86 | 0.85 | 0.86 | |
| | Reduction | 0.89 | 0.87 | 0.87 | 0.86 |
| | Stencil | N/A | N/A | N/A | |
| Graph2Par | Do-all | 0.88 | 0.87 | 0.87 | |
| | Reduction | 0.9 | 0.89 | 0.91 | 0.9 |
| | Stencil | N/A | N/A | N/A | |
| PROGRAML | Do-all | 0.92 | 0.90 | 0.91 | |
| | Reduction | 0.92 | 0.92 | 0.92 | 0.96 |
| | Stencil | 0.98 | 1 | 0.99 | |
| **PERFOGRAPH** | Do-all | 1 | 0.97 | 0.99 | |
| | Reduction | 0.97 | 1 | 0.99 | **0.99** |
| | Stencil | 1 | 1 | 1 | |

## 5.5 NUMA and Prefetchers Configuration Prediction

**Problem Definition:** An appropriate configuration of Non-Uniform Memory Access (NUMA) and hardware prefetchers significantly impacts program performance. In this experiment, we define the task of NUMA and prefetcher selection as predicting the right configuration within a given tuning parameter search space. We evaluate the performance of both PROGRAML and PERFOGRAPH for this task by converting each program in the dataset to PROGRAML and PERFOGRAPH graphs following the approach in [38].

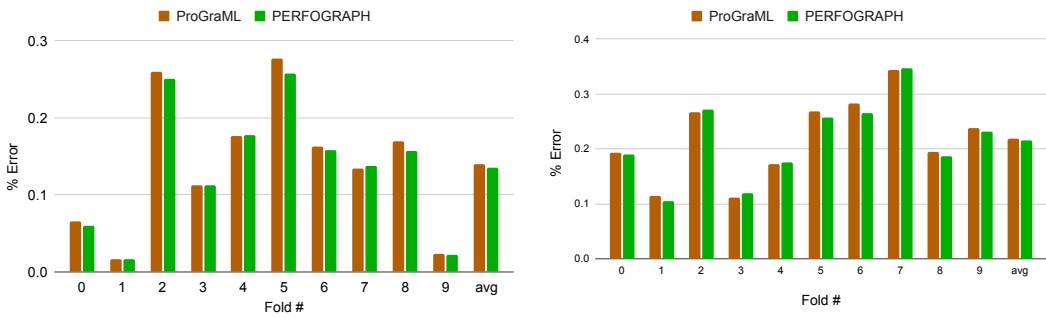

(a) Error distribution for Sandy Bridge architecture.    (b) Error distribution for Skylake architecture.

Figure 6: Breakdown of the NUMA and prefetchers configuration prediction per fold [lower is better].

**Dataset:** We use the dataset in [38], which includes a diverse set of intermediate representation files coupled with the optimal configuration [36]. The dataset incorporates various LLVM compiler optimization flags to produce different forms of the same program. There are 57 unique kernels (IR files) in this dataset, and around 1000 optimization flags are applied, resulting in 57000 IR files in total. Each IR file within the dataset is accompanied by its runtime on two architectures, Sandy Bridge and Skylake, across thirteen different NUMA and prefetcher configurations.

**Results:** Following the approach in the study of TehraniJamsaz *et al.*, we partition the dataset into ten folds for cross-validation. Figure 6a and 6b illustrate the performance results in terms of error rates. On average, PERFOGRAPH outperforms PROGRAML by achieving 3.5% and 1.8% better error rates on average for the Sandy Bridge and Skylake architecture, respectively. These improvements demonstrate the effectiveness of PERFOGRAPH compared to the state-of-the-art PROGRAML.

## 5.6 Thread Coarsening Factor (TCF) Prediction

**Problem Definition:** Thread coarsening is an optimization technique for parallel programs by fusing the operation of two or more threads together. The number of threads that can be fused together is known as the Thread Coarsening Factor (TCF). For a given program, the task is to predict the coarsening factor value (1, 2, 4, 8, 16, 32) that leads to the best runtime. The running time with coarsening factor 1 is used as the baseline for calculating speedups. For this task, we compare PERFOGRAPH against DeepTune [13], Inst2Vec [6] and PROGRAML [14]. The results of the baselines are quoted from [6]. Since PROGRAML has not been evaluated on this task in the past, we apply PROGRAML representation in our setup for comparison.

**Dataset:** We use the dataset of Ben-Nun et al. [13]. The dataset contains only 17 OpenCL kernels. For each kernel, the dataset has the runtime information on four different GPUs for the different thread coarsening factor values. Hence, for each kernel, we have the runtime corresponding to each thread coarsening factor value on a specific GPU device.

**Results:** we design the problem as a multi-class classification problem where, given a kernel, we try to predict which thread coarsening factor

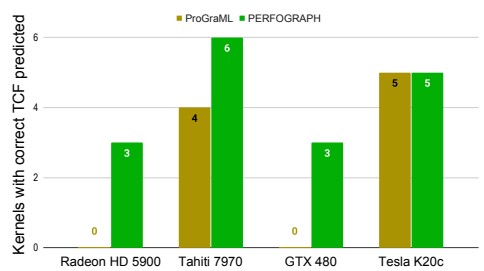

Figure 7: Correct TCF found by PROGRAML vs PERFOGRAPH [higher is better].

provides the highest performance. As the dataset is very small, we apply a 17-fold cross-validation approach. In each fold, we train our model on 16 data points, and the model is tested on the one unseen data point that is left out of the training set. Figure 7 shows the comparison of kernels with the correct Thread Coarsening Factor (TCF) found by PROGRAML and PERFOGRAPH. Across the four platforms in total PERFOGRAPH is able to correctly predict the TCF for 17 cases, whereas PROGRAML is able to find only 9 cases. In two of the platforms (AMD Radeon HD 5900 and NVIDIA GTX 480) where PROGRAML failed to find any kernel with the correct TCF, PERFOGRAPH can find three kernels in both of the platforms with the correct TCF value. As shown in 5, even though PERFOGRAPH outperforms PROGRAML on most computing platforms, it falls behind inst2vec. We posit the reason is that inst2vec has a pretraining phase where it is trained using skip-gram. On the other hand, 17 kernels are very small. Therefore, a DL-based model is not able to generalize enough. However, we can see that even on a smaller dataset, PERFOGRAPH achieved comparable speedups with respect to the current state-of-the-art models.

Table 5: Speedups achieved by coarsening threads

| Computing Platform | DeepTune | inst2vec | PROGRAML | **PERFOGRAPH** |
|---|---|---|---|---|
| AMD Radeon HD 5900 | 1.1 | **1.37** | 1.15 | 1.19 |
| AMD Tahiti 7970 | 1.05 | 1.1 | 1.00 | **1.14** |
| NVIDIA GTX 480 | **1.1** | 1.07 | 0.98 | 1.03 |
| NVIDIA Tesla K20c | 0.99 | **1.06** | 1.03 | 1.01 |

## 5.7 Algorithm Classification

**Problem Definition:** Previous downstream tasks showed that in most of the cases, PERFOGRAPH outperforms the baselines. Those tasks were mostly performance-oriented. We go further by applying PERFOGRAPH on a different downstream task, which is algorithm classification. The task involves classifying a source code into 1 of 104 classes. In this task, we compare the results of PERFOGRAPH to those of inst2vec, PROGRAML. The results of the baselines are quoted from [14].

**Dataset:** We use the POJ-104 dataset [30] in a similar setup as [14] that contains around 240k IR files for training and 10k files for testing.

**Results:** For this task, inst2vec has error rate of 5.17, whereas PROGRAML has error rate of 3.38. PERFOGRAPH yields an error rate of 5.00, which is better than inst2vec and slightly behind PROGRAML. One of the reasons is that PROGRAML already has a very small error rate in this task, leaving a very small gap for improvement; however still PERFOGRAPH's result is very close to that of PROGRAML. We could not reproduce the results in PROGRAML paper in our setup. When we applied PROGRAML in our setup, the error rate of PROGRAML was 6.00. Moreover, we posit that for algorithm classification, numbers are not a significant factor. Therefore, numerical awareness can confuse the models a little bit. However, this experiment shows that PERFOGRAPH is very close to PROGRAML's performance in this task and shows the applicability of PERFOGRAPH to a wider range of downstream tasks.

## 5.8 Ablation Study

We further analyzed how each one of the enhancements in PERFOGRAPH affects the end results. We performed an ablation study on the Device Mapping task and trained our GNN models on variations of PERFOGRAPH.

**Results without aggregate data type nodes:** First, aggregate data type nodes are removed from PERFOGRAPH representation. Please note that in this setup, the Digit Embedding is still applied. Table 6 and 7 shows the results. It can be seen that when the representation does not support aggregate data type, the error rate increases to 13% in AMD and 15% in the NVIDIA dataset. This clearly indicates that having aggregate-type nodes in the representation helped the model to learn the code features more accurately.

**Results without Digit Embedding:** Then Digit Embedding is removed from the pipeline for the second experiment and the aggregate data type nodes are kept. Table 6 and 7 shows the results. We can see that unlike having aggregate data type nodes, removing digit embedding does not hurt the error rate that much for the task of device mapping. However, we can still see a small increase (1.1%) in the error rate for the AMD dataset. For the NVIDIA dataset, the error rate increases from 10.0 to 10.6%.

Table 6: Summarizing PERFOGRAPH results for AMD device.

| Approach | Error (%) |
|---|---|
| DeepTune [13] | 28.1 |
| inst2vec [6] | 19.7 |
| PROGRAML [14] | 13.4 |
| PERFOGRAPH (without aggregate data type nodes) | 13.0 |
| PERFOGRAPH (without digit embedding) | 7.1 |
| PERFOGRAPH (aggregate data type nodes + digit embedding) | 6.0 |

Table 7: Summarizing PERFOGRAPH results for NVIDIA device.

| Approach | Error (%) |
|---|---|
| DeepTune [13] | 39.0 |
| inst2vec [6] | 21.5 |
| PROGRAML [14] | 20.0 |
| PERFOGRAPH (without aggregate data type nodes) | 15.0 |
| PERFOGRAPH (without digit embedding) | 10.6 |
| PERFOGRAPH (aggregate data type nodes + digit embedding) | 10.0 |

Finally, we can conclude that both components in our representation helped the model to learn the code features better to some extent. However, aggregate data type nodes in the embedding helped our model more than Digit Embedding for the task of device mapping. The reason can be that there are not many numbers in the dataset. However, in tasks where there are many numbers, Digit Embedding can play a significant role.

## 6   Conclusion and Future Work

In this paper, we presented PERFOGRAPH, an LLVM IR-based graph representation of programs that supports aggregate data types such as arrays and vectors and is numerical aware. Moreover, it addresses several limitations of the previous IR-based graph representations. PERFOGRAPH is evaluated on various downstream tasks, and experimental results indicate that PERFOGRAPH is indeed effective, outperforming state-of-the-art in most of the downstream tasks. PERFOGRAPH numerical awareness capability is limited to the numerical values that are available at the compile time. For future work, we intend to augment our representation by adding support for dynamic information and checking the possibility of integrating hardware performance counters with our representation. Moreover, we plan to develop a pre-trained embedding model using our representation. Having a pre-trained model will help to solve the problem of limited training samples in some downstream tasks.

## 7   Acknowledgement

This project was funded by NSF (#2211982) and Intel Labs. We would like to thank them for their generous support. Additionally, we extend our gratitude to the Research IT team[*] of Iowa

---

[*]https://researchit.las.iastate.edu/

State University for their continuous support in providing access to HPC clusters for conducting the experiments of this research project.

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
