# Supplementary Materials
# PERFOGRAPH: A Numerical Aware Program Graph Representation for Performance Optimization and Program Analysis

## 1 Insights of Digit Embedding

We investigated the effectiveness of Digit Embedding. Figure 1 shows the 2-d embeddings of integer numbers in the range [10, 60] and [100090-100140]. We take two ranges of numbers to better illustrate the results. We can see that the numbers in the (100090-100140) range are clustered together. The numbers with less difference, like (100133, 100134), (100127, 100128), and (100136, 100137), are close to each other. Also, the numbers with greater differences, like (100126, 100135) and (100196, 100133), are far from each other in the embedding space. A similar analysis is also true for the (10, 60) range. We can see that the numbers 21, 22, 28, and 29 are close to each other as they have small differences but numbers 11 and 59 are far from each other as they have greater differences.

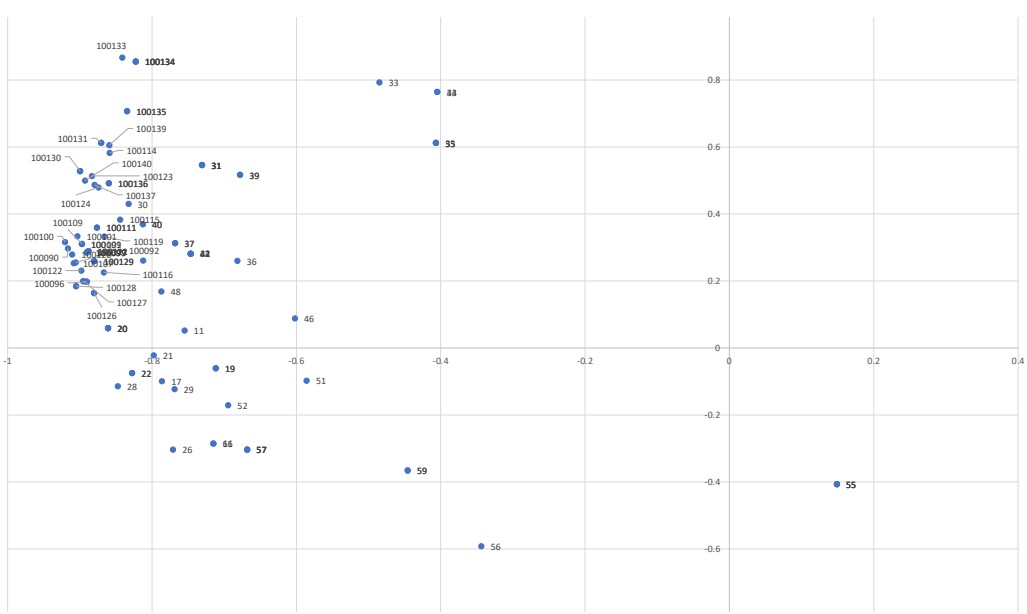

Figure 1: Embedding of integer numbers in the range [10-60] and [100090-100140]

We investigated with more ranges. Figure 2 shows the 2-d embedding of integer numbers in the range [1, 50] and [50000-500090]. Here we can also see that numbers with smaller differences like (50034, 50035, 50038, 50039), (13, 17), and (19, 21) are also closer to each other in the embedding space. Whereas numbers like (50011, 50028), (50017, 50029), and (2, 17) are far from each other in the embedding space as their differences are also greater.

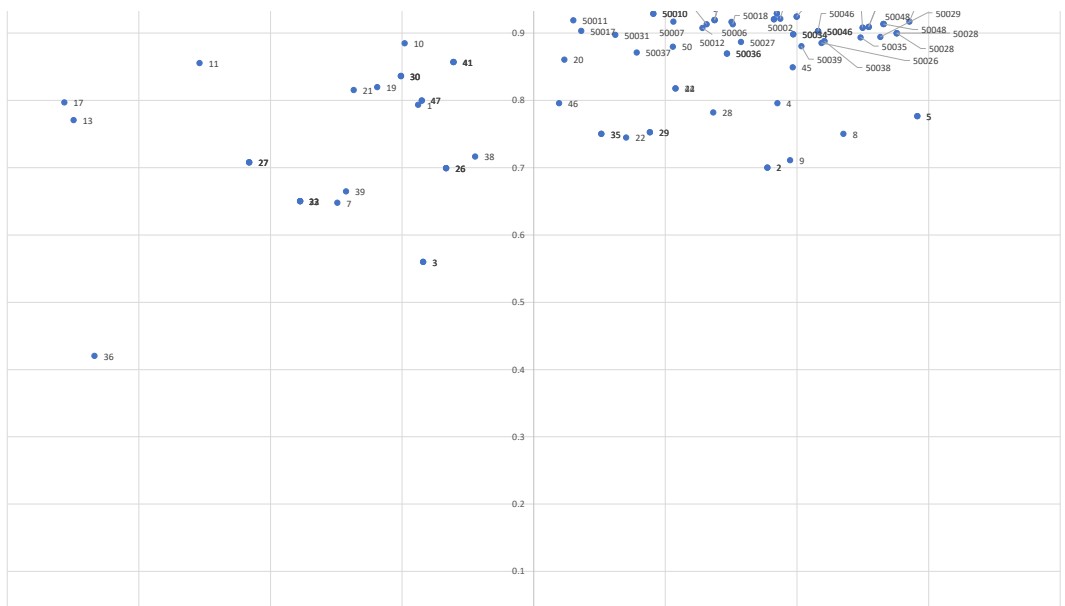

Figure 2: Embedding of integer numbers in the range [1-50] and [50000-500090]

Figure 3 shows the 2-d embedding of decimal numbers in the range [1.0, 10.0] and [20.0-31.0]. We can see that our embedding works similarly as the numbers with smaller differences like (2.236, 4.529), (1.647, 5.339), (23.0129, 23.3484, 24.5235, 25.8604) are close to each other in the embedding space. And the numbers with larger differences like (1.6478, 30.7010), (5.339, 30.5113) are far from each other in the embedding space.

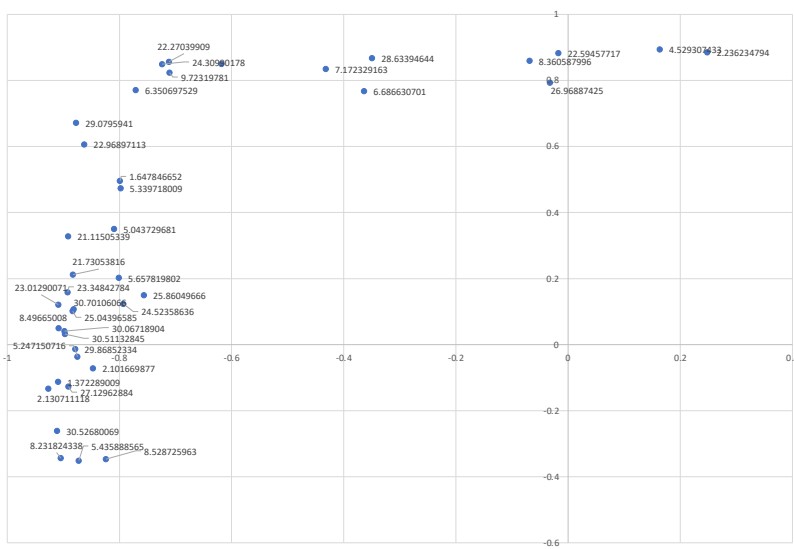

Figure 3: Embedding of decimal numbers in the range [1.0, 10.0] and [20.0-31.0]

So, the above examples clearly demonstrate the effectiveness of Digit Embedding for generating the embedding of both integer and decimal numbers.

## 2 Model's architecture

Table 1 shows the architecture and the hyper-parameters of our PERFOGRAPH model. For each one of the downstream tasks, we have two or more classes. While training the PERFO-GRAPH model, the class with the higher probability score is chosen as the predicted class and is compared against the actual class.

Figure 4 shows the error rate (loss value) of PERFOGRAPH model per epoch for the device mapping task. As shown, the model can learn from our PERFOGRAPH graph as it is able to decrease the error rate per epoch.

The source code of PERFOGRAPH for this task is available at the following link: `https://anonymous.4open.science/r/perfograph_devmap-532F/`

Table 1: PERFOGRAPH model architecture

| Parameter | Detail |
|---|---|
| Convolution Type | RGCN |
| # Conv Layers | 2 |
| Aggregation Function | Sum |
| Activation Function | Relu |
| Max Token Lenght | 40 |
| Embedding Dim | 120 |
| Padding | True |
| Hidden Dim | 64 |
| Output Layer Size | num_class |
| Optimizer | Adam |
| Learning rate | 0.01 (default) |

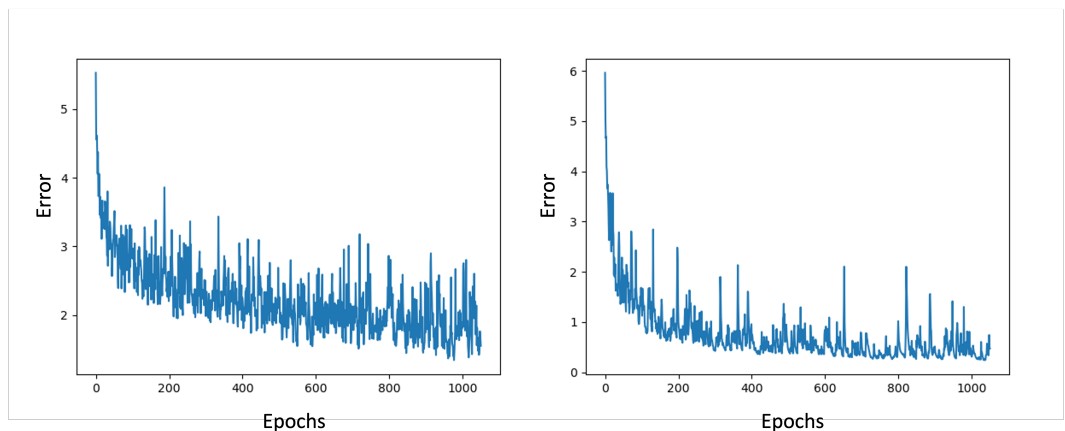

Figure 4: Error rate per epoch for AMD (left) and NVIDIA (right) datasets

## 3 Ablation Study

We further analyzed how each one of the enhancements in PERFOGRAPH affects the results. To this end, we performed an ablation study on the Device Mapping task, and training our GNN models on variations of PERFOGRAPH.

### 3.1 Results without Composite data type nodes

First, we remove the representation of composite data types in our PERFOGRAPH representation. Please note that in this setup, the Digit Embedding is still applied. Table 2 and 3 shows the results. We can see that when the representation does not support composite data types, the error rate increases to 13% in AMD and 15% in NVIDIA dataset. This clearly indicates that having composite-type nodes in the representation helped the model to learn the code features more accurately.

## 3.2 Results without Digit Embedding

We remove Digit Embedding from our pipeline for the second experiment and keep the composite nodes. Table 2 and 3 shows the results. We can see that unlike having composite-type nodes, removing digit embedding does not hurt the error rate that much for the task of device mapping. However, we can still see a small increase (1.1%) in the error rate for AMD dataset. For the NVIDIA dataset, the error rate increases from 10.0 to 10.6%.

Table 2: Summarizing PERFOGRAPH results for AMD device.

| Approach | Error (%) |
|---|---|
| DeepTune Cummins et al. [2017] | 28.1 |
| inst2vec Ben-Nun et al. [2018] | 19.7 |
| PROGRAML Cummins et al. [2020] | 13.4 |
| PERFOGRAPH (without composite data type nodes) | 13.0 |
| PERFOGRAPH (without digit embedding) | 7.1 |
| PERFOGRAPH (composite data type nodes + digit embedding) | 6.0 |

Table 3: Summarizing PERFOGRAPH results for NVIDIA device.

| Approach | Error (%) |
|---|---|
| DeepTune Cummins et al. [2017] | 39.0 |
| inst2vec Ben-Nun et al. [2018] | 21.5 |
| PROGRAML Cummins et al. [2020] | 20.0 |
| PERFOGRAPH (without composite data type nodes) | 15.0 |
| PERFOGRAPH (without digit embedding) | 10.6 |
| PERFOGRAPH (composite data type nodes + digit embedding) | 10.0 |

Finally, we can conclude that both components in our representation helped the model learn the code features better to some extent. However, composite data type nodes in the embedding helped our model more than Digit Embedding for the task of device mapping. The reason can be that there are not many numbers in the dataset. However, in tasks where there are many numbers, Digit Embedding can play an important role.

# 4   Details of Datasets:

## 4.1   Device Mapping

For this task, we used the Device Mapping Dataset. It contains around 256 OpenCL kernels. Around 671 IR files are extracted from these kernels. There are two types of devices: AMD and NVIDIA. For each of the devices, we have two classes: CPU and GPU indicating whether the kernel performs well in CPU or GPU. For AMD, we have 276 kernels for GPU and 395 kernels for CPU. For NVIDIA, we have 385 kernels for GPU and 286 kernels for CPU. We use 80% of the IR files for training, 10% for validation, and 10% for testing. For the AMD experiment, we used 36 IR files from CPU and 31 IR files from GPU for testing. For the NVIDIA experiment, we used 30 IR files from CPU and 37 IR files from GPU for testing.

## 4.2   Parallelism Discovery

For Parallelism Discovery, the OMP_Serial Dataset is used. The dataset contains 5731 compilable source c files. We compile these source files using Clang to create IR files. Also, 58 transformation flags from LLVM are applied to increase the dataset. The list of flags is provided in table 4 There are around 30k files in the training set. There are two classes: Parallel and Non-Parallel. The loops with the OpenMP pragma "#pragma omp parallel for" are considered as Parallel loops and the loops without this pragma are considered as Non-Parallel loops. To ensure the correctness of data labels, three existing parallelism suggestion tools: Pluto, autoPar, and DiscoPoP, are used to create three

Table 4: List of the transformation flags

| | |
|---|---|
| -adce | -dse |
| -always-inline | -aggressive-instcombine |
| -argpromotion | -lcssa |
| -bb-vectorize | -licm |
| -block-placement | -loop-deletion |
| -break-crit-edges | -loop-extract |
| -dce | -loop-extract-single |
| -deadargelim | -loop-reduce |
| -deadtypeelim | -loop-rotate |
| -die | -loop-simplify |
| -loop-unroll | -block-placement -break-crit-edges |
| -loop-unroll-and-jam | -break-crit-edges -argpromotion |
| -loop-unswitch | -break-crit-edges -dce |
| -lower-global-dtors | -dce -deadargelim |
| -loweratomic | -deadargelim -deadtypeelim |
| -lowerinvoke | -deadtypeelim -die |
| -lowerswitch | -die -dse |
| -adce -always-inline | -aggressive-instcombine -lcssa |
| -argpromotion -always-inline | -lcssa -licm |
| -bb-vectorize -argpromotion | -licm -loop-deletion |
| -loop-deletion -loop-extract | |
| -loop-extract -loop-extract-single | -loweratomic -lowerinvoke |
| -loop-extract-single -loop-reduce | -lowerinvoke -lowerswitch |
| -loop-reduce -loop-rotate | -lowerswitch -dse |
| -loop-rotate -loop-simplify | -die -dse |
| -loop-simplify -loop-unroll | -break-crit-edges -dce |
| -loop-unroll -loop-unroll-and-jam | -break-crit-edges -lower-global-dtors |
| -loop-unroll-and-jam -loop-unswitch | -dce -lowerinvoke |
| -loop-unswitch -lower-global-dtors | -deadargelim -loweratomic |
| -lower-global-dtors -loweratomic | |

testing subsets. So, all of the testing data are checked by at least one of the tools. The performance of PerfoGraph is reported for each of the testing subsets.

## 4.3 Parallel Pattern Detection

The OMP_Serial Dataset also contains source codes of three different patterns: Do-all (Private) (200 files), Reduction (200 files), and Stencil (300 files). The do-all and Reduction patterns are detected using DiscoPoP. For both Do-all and Reduction patterns 20 templates are extracted and then 10 different variations are applied to those templates. We consider simple variations like renaming variables/functions and changing operators to preserve the pattern of the original source code. There are currently no tools available for detecting Stencil patterns. So, they are labeled manually. There are three types of Stencils: 1-d, 2-d, and 3-d. For each type, we extracted 10 templates and applied 10 variations on each of those templates to generate the 300 Stencil loops. For generating the source codes from templates Jinja and SymPy are used. Some examples of templates and generated codes are shown in Listing 1, 2, 3, and 4. For more details regarding the dataset, it is encouraged to look into the paper by Chen *et al.* Chen et al. [2023].

```
for ({{cnt}} = 0; {{cnt}} <
{{limit}}; {{cnt}} = {{cnt}} +
{{constant}})
{
    //do-all operation
    {{operand}} = {{operand}}
    {{operator}} {{operand}};
}
```

Listing 1: A sample do-all template

```
for ({{cnt}} = 0; {{cnt}} <
{{limit}}; {{cnt}} = {{cnt}} +
{{constant}})
{
    /*reduction operation*/
    {{reduction_var}} = {{reduction_var}}
    {{reduction_operator}} ({{term}});
}
```

Listing 2: A sample reduction template

```
dst[0,0] @= src[0, 0] + src[1, 0] +
            src[-1, 0] + src[0, 1] +
            src[0, -1]
```

Listing 3: A sample Stencil template for Sympy input

```
for (int ctr_0 = 1; ctr_0 < 99;
        ctr_0 += 1) {
    double * RESTRICT _data_dst_00 =
    _data_dst + 100*ctr_0;
    double * RESTRICT _data_src_00 =
    _data_src + 100*ctr_0;
    double * RESTRICT _data_src_01 =
    _data_src + 100*ctr_0 + 100;
    double * RESTRICT _data_src_0m1 =
    _data_src + 100*ctr_0 - 100;
    for (int64_t ctr_1 = 1; ctr_1 < 99;
        ctr_1 += 1) {
        _data_dst_00[ctr_1] =
        _data_src_00[ctr_1 + 1]
        + _data_src_00[ctr_1 - 1]
        + _data_src_00[ctr_1]
        + _data_src_01[ctr_1]
        + _data_src_0m1[ctr_1];
    }
}
```

Listing 4: Generated stencil loop using Sympy

## 4.4 Numa and Prefetchers Configuration Prediction

The dataset we used for the Numa and Prefetchers Configuration Prediction is from a prior study by TehraniJamsaz *et al.* TehraniJamsaz et al. [2022]. It contains 57000 IR files generated by various LLVM compiler optimization flags. Each IR file within the dataset is accompanied by its runtime on two architectures, Sandy Bridge and Skylake, across thirteen different NUMA and prefetcher configurations.