# OpenReview forum: "PERFOGRAPH: A Numerical Aware Program Graph Representation for Performance Optimization and Program Analysis"
_NeurIPS.cc/2023/Conference — NeurIPS 2023 poster_

### Official Review · Reviewer_ubPp · 2023-06-28

**Soundness:** 3 good
**Presentation:** 4 excellent
**Contribution:** 3 good
**Rating:** 7
**Confidence:** 5

**Summary:**

The paper proposed a novel graph-based program representation, PerfoGraph, which is based on the current state-of-the-art method PrograML and aims to address its limitation by providing numerical awareness, introducing new tactics for handling local variables, and supporting aggregate data types. By conducting experiments on various performance-optimizing oriented downstream tasks, it is shown that the proposed PerfoGraph representation is more effective compared to the existing methods.

**Strengths:**

- The paper was clearly presented overall.
- The paper showed great originality in proposing a novel program representation method, PerfoGraph, that enhances previous results by providing numerical awareness, introducing new tactics for handling local variables, and supporting aggregate data types.
- Regarding numerical awareness, the authors introduced a novel idea to split the numbers into digits and positions before encoding, which could effectively represent numbers without a huge vocabulary.
- Regarding local variables, the author took good consideration of GNN's mechanism and adjusted the graph representation accordingly.



**Weaknesses:**

- Unconvincing experiment setup in comparison with PrograML.
In the original PrograML paper, MPNN was introduced as an encoder. However, the proposed PerfoGraph chose RGCN for the same role. The reviewer believes that it is possible to also use RGCN as PrograML's encoder. Thus, it is unclear whether the performance improvements in the comparisons with PorgraML stem from the novel graph representation or from a more suitable encoder.

**Questions:**

1. The authors claimed that the proposed PerfoGraph representation could better capture **composite** data structure information. However, the paper only introduced a novel method for incorporating list-like data structures, such as **arrays and vectors**, into the graph representation of the program. The reviewer's question is how PerfoGraph could support more general composite data types such as struct. From the current presentation, the reviewer's opinion is that the term **aggregate data types** suit better.
2. How does the choice of aggregation function impact the performance of PerfoGraph? Is it possible that the information compression introduced in the aggregation function confuses the deep learning model?
3. Do the authors consider moving the part of the ablation study from Supplementary Materials to the main text to better clarify the importance of numerical awareness in program representations?
4. Possible error in Figure 2a: In section 4.1, the authors used Figure 2a as an example of the PrograML representation to point out that PrograML assigns a position feature to the address edge of the store node. However, there is no corresponding label near the first store node.

**Limitations:**

The authors have addressed the limitations of their work in Section 6.

---

> ### Author Rebuttal · Authors · 2023-08-09
>
> - **(W1) Unconvincing experiment setup in comparison with PrograML:**\
> In the downstream tasks of sections 5.3, 5.4, 5.5, and 5.6, we used the RGCN encoder with ProGraML and compared it with PerfoGraph. For each of these downstream tasks, we used an RGCN model with the same architecture as described in Table 1 of Supplementary Materials both for PerfoGraph and ProGraML so that we can be sure that the differences in results are only because of changing the program representation. And as mentioned in the paper, results show that *PerfoGraph* representation performs better than *ProGraML* in almost all these tasks.
>
> - **(Q1) Aggregate data types:**\
> You are right. *PerfoGraph*, at the moment, only supports aggregate data types. Therefore, we will update the paper and will mention aggregate data types instead of composite data types.
> As explained to the question of inFa reviewer, *PerfoGraph* does not handle structs in a specific way different from *ProGraML*. In fact, *PerfoGraph* shows struct through a series of nodes. For example, a simple struct such as
>     ```
> struct {
>     int myNum;
>     int myAge;
>   } myInfo;
>   myInfo.myNum = 815;
>   myInfo.myAge = 23;
>     ```
> In LLVM IR is:
>     ```
> %2 = alloca %struct.anon, align 4
> %3 = getelementptr inbounds %struct.anon, %struct.anon* %2, i32 0, i32 0
> store i32 815, i32* %3, align 4
> %4 = getelementptr inbounds %struct.anon, %struct.anon* %2, i32 0, i32 1
> store i32 23, i32* %4, align 4
>     ```
> *PerfoGraph* will present these instructions by having\
> Control nodes as: `alloca`, `getelementptr`, `store`, `getelementptr`, `store`\
> Data nodes as: `%struct`, `i32 0`, `i32 1`, `i32 23`, `i32 815`\
> Which is basically the same way that *ProGraML* handles struct. We will update the paper and use the term **aggregate data types** to avoid confusion.
>
>
> - **(Q2) Does the choice of aggregation function impact the performance of PerfoGraph?**\
> We do observe some impact of aggregation function in the `GraphConv` layer of our GNN pipeline. We found `sum` aggregation provides the best performance overall. We did an experiment with the Device Mapping task using different aggregation functions.
>
>     | Device | Sum agr. accuracy  | Max agr. accuracy | Mean agr. accuracy | Min agr. accuracy  |
>     |--------|------|------|------|------|
>     | AMD    | 0.94 | 0.90 | 0.93 | 0.79 |
>     | NVIDIA | 0.90 | 0.85 | 0.90 | 0.84 |
>
>
> - **(Q3) Moving Ablations study to the main paper:**\
> As discussed in global repose, the ablation study was moved to the supplementary material file due to the lack of space in the paper. But for the final version of the paper, we will make sure to include the ablation study as requested. Thank you for pointing this out.
>
> - **(Q4) Possible error in Figure 2a:**\
> Thanks for pointing this out. The label is there but it is overlapping with the edge itself that's why it is difficult to see. We will fix it in the revised version of the paper.

---

> > ### Comment · Reviewer_ubPp · 2023-08-18
> >
> > Thank you for your rebuttal. It has further clarified my understanding of the proposed work.
> >
> > Given that I've previously given a 7-accept based on the novelty and originality of the proposed paper, I've chosen to keep my score unchanged.

---

### Official Review · Reviewer_vHZH · 2023-07-02

**Soundness:** 2 fair
**Presentation:** 2 fair
**Contribution:** 3 good
**Rating:** 7
**Confidence:** 4

**Summary:**

The paper presents a graph-based representation for LLVM IR programs for processing with graph neural networks. The representation is based on ProGraML, an established technique for such graph representations, but adds several features to the graph: collapsing nodes that refer to the same variable, adding additional edges between memory stores and the variables they can affect, explicitly representing constants in the graph, and representing compositional data types (e.g., arrays of arrays) as multiple distinct nodes rather than single nodes. The authors evaluate across a range of LLVM IR learning tasks, and show that Perfograph outperforms ProGraML (and all other baselines in most contexts).

**Strengths:**

* Finding and understanding good program representations is an important problem in the field of learning for code
* The proposed approach identifies a novel set of features which qualitatively seem important, and which are validated to be useful in the ablation study
* Assuming that the evaluation is indeed fair (see weaknesses below), the proposed approach significantly outperforms ProGraML, achieving a large increase in accuracy (seemingly solving many of these tasks near perfectly).


**Weaknesses:**

* Fairness of comparison against baselines:
  * Perfograph is a version of ProGraML with some added and removed nodes and edges. Each added node/edge adds additional power for a graph neural network. For a fair comparison against baselines (ProGraML in particular), the authors must show that for a given compute budget (i.e., FLOPs or wall clock time), Perfograph outperforms ProGraML; this would involve either scaling up/down ProGraML or Perfograph.
  * Are the datasets used exactly the same as in prior work? For example, Section 5.4 reports using an 80/20 train/test split, which I cannot find discussed in [10]
  * Line 372 mentions that the authors "could not reproduce the results in PROGRAML paper" for the algorithm classification task. Does this lack of reproducibility also affect the evaluated benchmarks that were not in the ProGraML paper (seemingly, everything other than 5.2)? If so, this is a major point that must be discussed at a high level in the paper.
  * Given the advent of large-context Transformers, I would be curious how well such a model performs when applied to the raw IR (i.e., without an explicitly featurized graph representation). However, since this paper is explicitly building on the prior work of ProGraML, I do not consider this to be a condition for acceptance.
* Overall, the clarity of the submission is moderate.
  * Section 4.1: without a deep understanding of the ProGraML representation, it is very hard to understand these graphs (and therefore the issues that the proposed approach fixes in them). Perhaps highlighting or otherwise identifying the specific nodes discussed could help.
  * Line 323: is this an arithmetic or geometric average?
  * Line 368: what are the units of these errors?
* Several claims about novelty or related work are misleading:
  * Lines 72-73: "this token-based representation... fails to capture..." This claim is misleading. Sequence processing models are entirely capable of learning such graph-structured relationships; such complex relationships are also not unique to code (e.g., natural language has pronouns, nested structures, etc.)
  * Line 176: the proposed "Digit Embedding" is very similar to some digit representations approaches used in Transformers (for example, Geva et al. 2020) which embed each digit, add/concatenate with a positional encoding, and aggregate with attention.
* I believe the paper would be significantly stronger with the ablation studies moved to the main body, but this is a stylistic choice and is not a factor in my score
* If I'm understanding Appendix 2 correctly, the Perfograph GNN is evaluated with only 2 layers; ProGraML is evaluated with 6 (Cummins et al., Section 6.2). How was this parameter chosen? What is the average diameter of the evaluated graphs (for both Perfograph's graphs, and ProGraML's graphs)?


**Questions:**

* Please see the questions in "clarity" in the discussion of weaknesses above (and other questions throuhgout).
* Does Perfograph use more compute or have more capacity than ProGraML? Does it still outperform ProGraML when given equal power?
* What steps have the authors taken to ensure that the implementation of ProGraML compared against in Sections 5.3-5.7 is correct (since as noted in Line 372, in one context the authors "could not reproduce the results in PROGRAML paper")?
* Do all other baselines use the exact same training/test set split?
* How were hyperparameters tuned for Perfograph?

I would be willing to raise my score given evidence that for the same compute budget, Perfograph still outperforms ProGraML (or evidence that my assumption here, that Perfograph uses more compute than ProGraML, is incorrect).


**Limitations:**

The paper does not explicitly discuss limitations of the technique.

---

> ### Author Rebuttal · Authors · 2023-08-09
>
> - **(W1.a, Q2) The computation cost PerfoGraph:**\
> We conducted an experiment in terms of the time it takes to train the GNN model using *PerfoGraph* versus *ProGraML*. We found out that *PerfoGraph* takes less training time than *ProGraML*, and *PerfoGraph* also has better performance than *ProGraML*. For more details on this experiment, please refer to the global response. Thank you.
>
> - **(W1.c, Q3) Lack of reproducibility of ProGraML for algorithm classification:**\
> Apologies for any ambiguity. As detailed in Section 5.7, we assessed the algorithm classification performance using both *ProGraML* and *PerfoGraph*. However, our replication did not mirror the error rate mentioned in the *ProGraML* paper. We observed an error rate of 6%, whereas the *ProGraML* paper cited 3.38%. This minor discrepancy in error rate could potentially be rectified with further fine-tuning of the GNN model tailored for *ProGraML*. Nonetheless, the *ProGraML* repository did not provide any checkpoints. For this paper, we initialized the GNN models with identical settings and gauged the performance of *ProGraML*. In the interest of fairness, we have presented both the error rate from the *ProGraML* paper and our results.\
> For the Device Mapping task (section 5.2), we were able to reproduce the same results as reported in the *ProGraML* paper. So, we compared *PerfoGraph* with the accuracy mentioned in the *ProGraML* paper.\
> Other downstream tasks (sections 5.3, 5.4, 5.5, 5.6) are not described in the *ProGraML* paper. For a fair comparison of *PerfoGraph* with *ProGraML* for these downstream tasks, we keep everything the same in our pipeline except for the program representation.
>
> - **(W2.a) The difference with ProGraML in Figure 2:**\
> We will make sure to highlight those nodes and edges in Figure 2 for better visibility. Some enhancements of PerfoGraph are shown in the global response pdf file.
>
> - **(W2.b) Type of average on line 323 (Arithmetic of Geo Mean?):**\
> This is an arithmetic average. We ensure to clarify this in the paper.
>
> - **(W2.c) Unit of errors on line 368:**\
> We represent the error rates as percentages. For example, inst2vec has an error rate of 5.17 means inst2vec has an accuracy of 94.83%. We will include the percentage sign in the revised submission to avoid confusion.
>
> - **(W3.a) The claim of token-based representation fails to capture relations is misleading:**\
> Sorry for the confusion. While there have been studies that incorporate relations in token-based representations, these often necessitate specialized models, tailored representations, or additional training data and time to effectively capture relation information. In contrast, previous works (Zhang, Jian, et al. 2019, and Chen, Le, et al. 2023) have shown that graph representations of code inherently encapsulate critical relational data more naturally than their token-based counterparts. Using compiler-generated IR, *PerfoGraph* ensures that relational information is both accurate and precise, making it an efficient representation for models to capture code features. We will adjust our description of token-based representation to ensure clarity.
>
> -  **(W3.b) The Digit Embedding is similar Geva et al:**\
> Thank you for pointing this out. The idea of digit embedding was actually inspired by input and position embeddings in Transformers. We applied the position encoding to each one of the digits of a number. The idea can be considered similar to character encoding with assigning a position to each character. However, to the best of our knowledge, we did not find any tool or model that can do this type of embedding for numbers. Hence we proposed Digit Embedding.
>
> - **(W4) Moving the ablation to the main paper:**\
> Thank you for mentioning this point. As mentioned in the global response, we will make sure to include the ablation study in the paper.
>
> - **(W5) What is the diameter of the graphs:**\
> We measured the average diameter of the 30k training IR files in the Parallelism Discovery task as an example. Below are the results:\
> Avg diameter of PerfoGraph:  28.3\
> Avg diameter of ProGraML: 31.54
>
>
> - **(W1.b, Q4) Is Train/Test Split in other baselines the same?**\
> Yes, the datasets are used exactly the same as in prior works.\
> Sec 5.2: Same as *ProGraML* paper. We do 80% training, 10% validation, and 10% testing\
> Sec 5.3: Same as *Graph2Par* paper. Around 30k for training and testing on three subsets *Pluto* (4032 IR files), *autoPar* (3356 IR files), and *DiscoPoP* (1226 IR files)\
> Sec 5.4: *Graph2Par* paper reproduced the *Pragformer* work to show the comparisons, and in the *Pragformer* repository, they used 80% and 20% split. So we used the same 80% training and 20% testing split ratio\
> Sec 5.5: Same as in the study of TehraniJamsaz et al., 10-fold cross-validation\
> Sec 5.6: Same as in the *inst2vec* paper, leave one out cross-validation\
> Sec 5.7: Same as *ProGraML* paper. 240k IR for training and 10k IR file for testing.
>
> - **(W5,Q5) How are hyperparameters tuned?**\
> We have tuned the hyperparameters of the model experimentally. We did experiment with a higher number of layers; however, it did not bring any benefits, despite the higher computation cost. Below we show the comparison of accuracies among different layers of our GNN-based model for the task of Device Mapping.
>
>     | Device | 2 layers accuracy  | 3 layers accuracy | 4 layers accuracy | 5 layers accuracy  | 6 layers accuracy|
>     |--------|------|------|------|------|------|
>     | AMD    | 0.94 | 0.88 | 0.91 | 0.87 | 0.90 |
>     | NVIDIA | 0.90 | 0.85 | 0.90 | 0.90 | 0.88 |
>
>     We use Adam optimizer as it is the default configuration for DGL-based RGCN and is widely used (e.g., in the ProGraML paper). We use Relu activation (default of DGL-RGCN) as it is also widely used. We experimented with commonly used hidden layer sizes (32, 48, and 64) and learning rates (0.1, 0.01, and 0.001) and observed best results are obtained with hidden layer size 64 and learning rate 0.01.

---

### Official Review · Reviewer_inFa · 2023-07-07

**Soundness:** 3 good
**Presentation:** 4 excellent
**Contribution:** 3 good
**Rating:** 6
**Confidence:** 3

**Summary:**

This work proposes Perfograph, a program graph representation based on LLVM-IR and an extension to ProGraML. This graph representation is designed for the purpose of performance optimization and program analysis applications based on graph neural networks (GNN). This work made three contributions to the existing representation. First, it tracks reused local identifiers and memory locations. Second, it uses decimal-based encoding to embed numerical constants into the graph. Last, it breaks down array and vector types into multiple nodes. Perfograph is tested on 6 downstream tasks: device mapping, parallelism discovery, parallel pattern detection, NUMA and prefetchers configuration prediction, thread coarsening factor (TCF) prediction, and algorithm classification. In the device mapping challenge, the new model outperforms ProGraML by 7.4% on the AMD dataset and 10% on the NVIDIA dataset.

**Strengths:**

1. This paper is well organized.
2. This work clearly listed all 3 of its design decisions.
3. This work contains detailed evaluations on multiple targets.
4. Each evaluation contains a carefully described problem definition, dataset, and results.

**Weaknesses:**

1. The improvement of this work over ProGraML seems limited. For example, the 3 main design contributions are not fundamentally new. They are more like minor tweaks on the existing ProGraML system.
2. The evaluation section does not show how each design choice contributes to the overall improvement.

**Questions:**

1. It claims that “Perfograph is built on top of ProGraML” in Section 4, but also claims “A **new** compiler and language agnostic program representation based on LLVM-IR”. Is this work an extension of ProGraML, or is it a new system? This is somewhat unclear.
2. Section 4.1 claims that this work combines multiple writes to the same variable into one node. However, LLVM deliberately separates multiple writes for the ease of certain aliasing analyses, the opportunities of reordering, and optimizing for parallelism. It would be helpful to provide more evidence that this combined representation outperforms separated representation in all (or most) downstream tasks.
3. Section 4.2 mentions that decimal-based encoding is used for numerical constant embedding. This seems counterintuitive since computers natively speak binary.
4. Section 4.3 claims that Perfograph supports composite data types (but only covers arrays and vectors). It would be helpful to further explain how Perfograph deals with programs that use structs (and maybe unions).

---

> ### Author Rebuttal · Authors · 2023-08-09
>
> - **(W1) Differences and Contributions that differentiate PerfoGraph from ProGraML:**\
> While we acknowledge the similarities between *PerfoGraph* and *ProGraML*, as both represent programs as graphs using LLVM Intermediate Representations, *PerfoGraph* differs itself from *ProGraML* by:\
> A) A more precise representation of local variables\
> B) Incorporating and embedding numbers\
> C) Supporting aggregate data types.\
> Moreover, PerfoGraphs enhancements are not dependent on ProGraML. For instance, numerical embedding and our aggregate data types representation can easily be incorporated into other graph representations.
>
> - **(W2) Contribution of each design choice:**\
> Thank you for mentioning this. To provide insights into the contribution of each design choice, we have conducted an ablation study. For instance, for the Device Mapping dataset removing the composite data types increased the error rate by 13% for the AMD dataset. To see all the details on the ablation study, please refer to the global response and the supplementary materials file. Thank you.
>
> - **(Q1) Is this work an extension of ProGraML, or is it a new system?**\
> We drew inspiration from *ProGraML* but observed certain limitations in its design. Specifically, *ProGraML* sometimes fails to capture some essential information in its representation, notably numbers, composite data type nodes, and local identifiers. In *PerfoGraph*, we prioritized refining the representation to address these shortcomings. Given that ProGraML integrates foundational graphs like CFG and DFG when using IR as the basic element for graph construction, it is challenging to bypass it entirely. Our approach in *PerfoGraph* sets itself apart by offering novel digital embedding for numbers and tailored representations for data type nodes and local identifiers.
>
> - **(Q2) Combining multiple writes to the same variable:**\
> You are indeed right that LLVM separates the multiple writes to the same variable for the sake of simplifying optimization choices. In our representation, we do not combine multiple writes to the same node. Sorry for the confusion. We make sure to clarify this in the paper. For the example in Figure 2a in the paper, there are two `store` instructions. The first `store` instruction corresponds to assigning 0 to the variable *i*, and the second `store` instruction corresponds to increasing the value of *i* by 1. In LLVM IR, this is shown as follows:
>     ```
> store i32 0, i32* %2, align 4
> store i32 %4, i32* %2, align 4
>     ```
> As we can see, the two `store` instructions are writing to the same local identifier, which is `%2`. However, ProGraML creates a separate node to present `%2` in each `store` instruction, as we can see in Figure 2a. This will make it difficult for DL models to understand that those `store` instructions are writing to the same local variable. In contrast, *PerfoGraph* considers only one node to represent `%2`.  For each temporary variable (`%2`, `%3`, …) that LLVM creates, *PerfoGraph* will also create separate nodes. The issue with ProGraML was that it was creating several nodes for the same temporary variable (Figure 2a).
>
>
> - **(Q3) Numerical Embedding is counterintuitive since computers speak binary:**\
> We proposed Digit Embedding to enable DL/ML models to be aware of numerical values. That is, if, for example, the loop bound (which can be a number) is known at compile time, we want the DL model to understand the scale and value of the number without losing its generalizability, that is, facing unknown numbers at test time. Digit embedding will essentially help to generate embedding for numeric tokens, which are present in the nodes of the graph representation. This will help DL models to make better predictions. Moreover, we intend to make the DL model aware of the numerical values at compile time. Therefore, we assume at the LLVM IR phase, we are not dealing with binary code.
>
> - **(Q4) Support for struct:**\
> Currently, *PerfoGraph* does not handle structs in a specific way different from *ProGraML*. In fact, *PerfoGraph* shows struct through a series of nodes. For example, a simple struct such as
>     ```
> struct {
>     int myNum;
>     int myAge;
>   } myInfo;
>   myInfo.myNum = 815;
>   myInfo.myAge = 23;
>     ```
> In LLVM IR is:
>     ```
> %2 = alloca %struct.anon, align 4
> %3 = getelementptr inbounds %struct.anon, %struct.anon* %2, i32 0, i32 0
> store i32 815, i32* %3, align 4
> %4 = getelementptr inbounds %struct.anon, %struct.anon* %2, i32 0, i32 1
> store i32 23, i32* %4, align 4
>     ```
> *PerfoGraph* will present these instructions by having\
> Control nodes as: `alloca`, `getelementptr`, `store`, `getelementptr`, `store`\
> Data nodes as: `%struct`, `i32 0`, `i32 1`, `i32 23`, `i32 815`\
> Which is basically the same way that *ProGraML* handles struct. We will update the paper and use the term **aggregate data types** to avoid confusion, as suggested by another reviewer.
> Thank you for pointing this out.

---

> > ### Comment · Reviewer_inFa · 2023-08-21
> >
> > Many thanks for the authors' detailed response. The new performance data and technical details are very helpful for me to better understand this work. Although I still have some minor concerns about the novelty of this work due to the similarities between PerfoGraph and ProGraML, I would like to change my score to "Weak Accept" because I believe this work contains enough intellectual merits.

---

### Official Review · Reviewer_f7Gk · 2023-07-07

**Soundness:** 3 good
**Presentation:** 4 excellent
**Contribution:** 3 good
**Rating:** 6
**Confidence:** 4

**Summary:**

The research identifies limitations in the current state-of-the-art program representation PROGRAML, in capturing features of numerical values and composite data types.
To address these limitations, this work introduces an enhanced GNN-based program representation for LLMV-IR with modifications to the nodes and edges of the program graphs to capture features of numerical values and composite data structures. It also presents a digit embedding approach to enhance numerical awareness within the representation.
The evaluation shows that PERFOGRAPH reduced the prediction error rate by 7.4% (AMD dataset) and 10% (NVIDIA dataset) in the Device Mapping challenge compared to PROGRAML. Additionally, it outperforms PROGRAML and traditional rule-based methods in tasks such as determining if a loop is parallelizable, classifying the parallel patterns, and configuring NUMA and prefetchers selections.


**Strengths:**

1. PERFOGRAPH employs effective graph modifications to enhance the learnability of the LLVM-IR graph for GNNs. These methods are well-reasoned and provide valuable domain information for improving program graph abstractions in learning.
For instance, PERFOGRAPH unifies the graph nodes for the same local identifier variable in the program graph. It also modifies the LLVM IR graph to add edges from the store instruction to the variables it modified. These modifications all simplify the complex structure of the LLVM-IR graph.

2. PERFOGRAPH also distinguishes between composite data structures and regular data types, which prior approaches did not adequately address. This distinction is crucial because operations involving composite data structures can differ significantly from those involving regular data types.

3. The evaluation of PERFOGRAPH is comprehensive, considering six performance-oriented tasks. The results demonstrate that PERFOGRAPH outperforms PROGRAML in terms of accuracy on these tasks. More importantly, in parallelism discovery tasks, PERFOGRAPH achieves higher accuracy compared to traditional rule-based approaches.


**Weaknesses:**

1. It is unclear to me how the digit embedding ensures that unknown numbers are not encountered during the test phase and what specific information is encoded through the addition of digit and position embeddings.
2. The reasons behind representing multi-dimensional arrays as multiple nodes are not adequately explained. It remains unclear how this representation enhances expressiveness or facilitates reasoning by GNNs.
3. An ablation study that isolates the individual benefits of LLVM-IR graph modification, digit embedding, and support for composite types is absent. Such a study would provide valuable insights into understanding the specific advantages offered by each component.
4. PERFOGRAPH may have higher computational requirements compared to PROGRAML, potentially posing challenges for its practical deployment.



**Questions:**

1. It would be helpful to visualize the digit embedding values of different numerical values to facilitate better understanding.
2. It would be beneficial to compare the cascaded context approach for representing multi-node representation with alternative representations (e.g. storing only the size of one rank) to assess the effectiveness in encoding array information.
3. To provide a more comprehensive analysis, I recommend conducting an ablation study for each major representation enhancement to understand the specific benefits of each component.
4. It would be valuable to measure and compare the inference time of PERFOGRAPH with PROGRAML and other rule-based analysis methods to evaluate the computational efficiency.

---

> ### Author Rebuttal · Authors · 2023-08-09
>
> - **(W1) Unknown numbers during the testing phase:**\
> Here, by unknown numbers, we mean numeric tokens not encountered by the model during the training phase but present in the testing phase. Digit Embedding allows us to generate embedding for those unknown numbers. Because it breaks the numbers into digits, which are from 0 to 9 only. Then we get the numeric position of each of the digits. Then embedding is generated by combining each digit with its numeric position. This way, we can generate embedding for unknown numeric tokens. The most important feature of Digit Embedding is that it uses digits and their positions to generate embedding rather than the number itself, and as we know, digits are from 0 to 9.
>
> - **(W2, W3, Q1, Q2, Q3) Ablation Study, the Benefits of representing composite data types and visualization of Digit Embedding values:**\
> We have conducted an ablation study to have more insights on the contribution of each one of the enhancements that *PerfoGraph* brings; for example, in terms of composite data types, removing them increases the error rate of the AMD dataset by %13 for the device mapping dataset. Please refer to the global response and supplementary material file for more details. We have also included the visualizations of Digit Embedding values in the supplementary material file.
>
> - **(W4, Q4) The computation cost of PerfoGraph:**\
> Thank you for pointing this out. As mentioned in the global response, our observations indicate that *PerfoGraph* does not impose significant overhead when training DL models. In fact, we observed a slightly lower overhead with *PerfoGraph*. Please refer to the global response for more details regarding the computation time experiments.

---

> ### Comment · Reviewer_f7Gk · 2023-08-19
>
> Thank the authors for the response. I found the supplementary material answered most of my questions.
>
> The ablation study indicates that digital embedding has a lesser impact on accuracy (1%) compared to the composite data type (>5%). The title is a bit misleading in this case as it emphasizes more on numerical awareness.
>
> I'm still not fully convinced that digital embedding is effective in capturing the magnitude of numerical values. One observation is that it seems it will cluster numbers with the same digits together (e.g. the point of number 50 is very close to the numbers in [50000-50090] in Fig.2 of the supplementary material.
>
> I tend to keep my score unchanged.

---

> > ### Author Response · Authors · 2023-08-21
> >
> > Thank you for sharing your feedback. We acknowledge that a slightly reduced impact of Digit Embedding is observed on the Device Mapping dataset, However as this data set is relatively small, we conducted a further experiment on the DiscoPoP subset of the Parallelism Discovery task by eliminating Digit Embedding. We found our accuracy dropped to **95.26% (3.74% decrease)** with Digit Embedding removed from that experiment. From this observation, we believe it is fair to say that the role of Digit Embedding can be more prominent on more extensive datasets.

---

### Author Rebuttal · Authors · 2023-08-09

We would like to thank all reviewers for their constructive feedback and comments.
Hereby, we address some of the common concerns.
- **Moving the Ablation Study to the main paper:**\
Due to the lack of space in the paper, we currently have the ablation study results in the Supplementary file. We will make sure to move and include the ablation study in the main paper. Thank you.
- **Ablation Study Results:**\
We have conducted experiments on the contribution of each design choice and presented the results as Ablation Study in the Supplementary Materials file.
For example, for the device mapping problem, removing composite data type nodes increases the error rate to %13 for the AMD dataset and 15% for the NVIDIA dataset. The supplementary material file contains more information about the Digit Embedding, Architecture of the models, Ablations study, and dataset.
- **Computation cost of PerfoGraph:**\
Thank you for pointing this out.
We measured the computation cost of *PerfoGraph* against *ProGraML*. For this experiment, we took the *OMP_Serial* dataset of the Parallelism Discovery task, which contains 30k IR files for training. The hardware configuration is the same as reported in section 5.1. We train our GNN model for 100 epochs with both *ProGraML* and *PerfoGraph* five times and report the average training time as follows:\
*PerfoGraph: 14 minutes and 59 seconds.*\
*ProGraML: 15 minutes and 56 seconds.*\
Then we measured the testing time with both *ProGraML* and *PerfoGraph* representation for the *Pluto* subset of *OMP_Serial* dataset as it is the largest subset containing 4032 test IR files. For measuring testing time, we again measured the testing time five times and reported the average testing time.\
The average testing time results for  the *Pluto* subset are as follows:\
*PerfoGraph: 881 milliseconds*\
*ProGraML: 1231 milliseconds*\
And as already shown in the paper, for the *Pluto* subset, *PerfoGraph* has higher accuracy **(91%)** than *ProGraML* **(89%)**.
Moreover, we measured the average diameter of *PerfoGraph* and *ProGraML* for the 30k IR files in the Parallelism Discovery training dataset:\
*Average diameter of PerfoGraph:  28.3*\
*Average diameter of ProGraML: 31.54*\
Based on these observations, we believe it is fair to say that the computational overhead of *PerfoGraph* is slightly less than *ProGraML*, while *PerfoGraph* has shown better performance, as discussed in the paper.

---

### Decision · Program_Chairs · 2023-09-21

**Decision:**

Accept (poster)

**Comment:**

This is a primarily empirical submission, presenting three modifications to the existing ProGraML approach to representing LLVM IR as graphs and illustrates their usefulness with experiments on a range of a performance-oriented tasks from the literature, showing that Perfograph autperforms baseline methods.

The proposed modifications are small and not grounded in theory, but instead stem from observations of shortcomings in ProGraML (as highlighted by reviewer inFa). This implies a high quality bar for the experimental section of the paper. As noted above, these experiments are wide in nature, covering six different tasks. However, the considered baselines are fairly dated (apart from Graph2Par and Pragformer used in Sect. 5.3 and 5.4, all were published in 2020 or earlier) and do not reflect that in the wider ML4Code community, LLMs have fully replaced graph-based models (see e.g. https://openreview.net/pdf?id=B1lnbRNtwr for a paper from 2020 on this). This was raised by reviewer vHZH, and the authors chose not to respond to this in their rebuttal.

Indeed, nearly-concurrent work by https://arxiv.org/pdf/2306.14979.pdf by Chen et al (the authors of reference [10] in the paper, from which the tasks in Sect. 5.3 and 5.4 are drawn) illustrates the importance of a comparison to LLMs: gpt-3.5-turbo with retrieval-augmented prompts outperforms all considered methods on parallelism detection (Sect. 5.3 in the submission), including the Graph2Par baseline compared to in this submission. This is in line with other recent works, which show that LLMs succeed on harder, more free-form performance optimization tasks (e.g. https://arxiv.org/pdf/2206.13619.pdf and the very recent https://arxiv.org/pdf/2306.17281.pdf).

Finally, the paper in its submitted form does not include substantial ablation studies. This point was raised by reviewers f7Gk, inFa; reviewers vHZH and ubPp found some unreferenced experiments in the supplement and asked for them to be moved into the main paper. While the authors have promised to include them in the main text, these ablations remain incomplete: they do not consider the effect of one of the proposed changes (Sect. 4.1) at all.

To summarise, this submission proposes small improvements to the existing framework ProGraML framework and illustrates their utility on six benchmarks from the literature. However, it does not does not fully ablate the proposed changes and does not engage with the recent dominant trend in the research community (to be clear, I am not saying that every paper needs to follow this trend, but you have to acknowledge such things and compare performance).

As AC, I found this submission to be by far the hardest decision to make, and I spent significant time on investigating the paper and ended up doing a full review (see below for detailed comments). In the end, and given the limited space at a very competitive conference such as NeurIPS, I believe that the submission is not ready to be accepted: to fully illustrate the usefulness of the proposed techniques, wider experiments taking LLMs into account are required. LM4HPC (https://arxiv.org/pdf/2306.14979.pdf) illustrates how this could be done.

In final discussions with the SAC and PC, the issue of fairness was raised: the authors had no opportunity to rebut the weaknesses highlighted in this meta-review, and so the decision was made to accept the paper, even though there are open issues that should be addressed. The authors are encouraged to improve the final version of the paper in response to these comments.

Detailed comments from an in-depth review:
* l35-38: "[ProGraML's] limitations stem from neglecting numerical values available at compile time and the inadequate representation of composite data types." - this statement is just foreshadowing what the paper proposes, but is not actually supported by evidence that these are (all of) the most important limitations.
* l47-48 (contribution 1): "A new compiler and language agnostic program representation" - as noted by reviewer inFa, this is overclaiming a bit, given that the paper is largely based on ProGraML (and even defers to it for exact definitions).
* l73-75: "[token-based representation] fails to capture the unique relationships and dependencies between different elements of the code, which can limit its effectiveness in tasks such as compiler optimization and code optimization" - no evidence provided. The cited papers (e.g., [22]) already show by-now smallish LMs (<500M parameters) being on par with graph-based approaches, and follow-ups such as https://arxiv.org/pdf/2305.07922.pdf show pure LMs outperform graph-based approaches. See the discussion above for why this is problematic.
* Sect. 4.1: I found the writing in this section highly confusing - it would be very helpful if the LLVM IR that forms the basis of the graph would be presented explicitly, as it is repeatedly referred to. (I ended up using clang to generate it myself, which should not be expected of readers)
* Sect. 4.1: I'll also note that the use of edge features to indicate argument positions is a commonly used trick (though usually not highlighted). The earliest occurrence I could find is https://aclanthology.org/P17-1105.pdf.
* Sect. 4.2: The proposed "digit embedding" is a variation of standard schemes known from NLP (where embedding numbers by digit is common, and a natural consequence of BPE approaches). The core novelty here is the positional encoding of floating point numbers. However, positional embeddings are commonly used in combination with attention mechanisms. The aggregation functions proposed here (min/max/mean) seem to be unable to make use of this information, as they are order-invariant. In particular, when using the mean, the "positional" part of the embedding is entirely dependent on the length of the number (as `avg([emb(digit1) + emb(pos1), ..., emb(digitN) + emb(posN)])` can be rewritten as `avg([emb(digit1), ..., emb(digitN)]) + avg([emb(pos1), ..., emb(posN)])`).
Consequently, pairs of numbers such as `91` and `19` or `999.111` and `111.999` would yield the same representation.
Note that the paper does not actually provide any details on which aggregation function is used for the experiments, but in Fig. 1 of the supplementary material, exactly this effect shows up (e.g. for 34/43, 35/53). I'll note that the plot is surprising in that some numbers seem to be missing entirely (12, 13, 14, ...), as this would have made this issue very visible.

* Sect. 4.3: Again, the description here is somewhat confusing. The paper does not precisely define node representations, and it's unclear how the "additional features" are used (array length & data type). We have to assume that this is addition to the ProGraML node features (simple embedding lookup of the full textual node representation). I found the choice of using several chained nodes surprising (given that the authors raise the issue of the number of hops required to "understand" a construct in Sect. 4.1) and would appreciate more discussion of alternative representations here (e.g., you could imagine a similar scheme as in Sect. 4.2, where an array of shape [d1, ..., dN] and type T is represented by a embedding [d1, ..., dN] with additional positional information and concating that with an embedding of T).

* Sect. 4.2/4.3: Both problems tackled here (a naive representation of a potentially large set of node labels) would likely be solved by a more advanced approach of representing node labels, as discussed by http://proceedings.mlr.press/v97/cvitkovic19b/cvitkovic19b.pdf.